# Screening Accuracy of FeNO Measurement for Childhood Asthma in a Community Setting

**DOI:** 10.3390/children9060858

**Published:** 2022-06-08

**Authors:** Kamil Barański, Jan Eugeniusz Zejda

**Affiliations:** Department of Epidemiology, Poland Faculty of Medical Sciences in Katowice, Medical University of Silesia in Katowice, 40-752 Katowice, Poland; jzejda@sum.edu.pl

**Keywords:** nitric oxide, asthma, children, epidemiology

## Abstract

(1) Background: The exhaled fractional nitric oxide is a well-recognized biomarker used in clinical settings for controlling and managing asthma. Less is known about the value of Fractional Exhaled Nitric Oxide (FeNO) measurement in epidemiological studies on childhood asthma, although available evidence suggests that an increased FeNO is associated with an increased risk of asthma. (2) Aim: The aim of the study was to assess FeNO accuracy in the identification of children with asthma, participants in a population-based respiratory survey. (3) Material and methods: The cross-sectional study included 449 children, 224 (49.9%) boys and 225 (50.1%) girls aged 6–10 years. The FeNO was measured in 449 children; Spirometry tests were completed with 441 children, but technically acceptable spirometry was done in 350. All participants fulfilled the questionnaire (ISAAC) for assessment of the status of their respiratory system on which diagnosis was based on. FeNO and Spirometry were performed according to ERS/ATS recommendations. (4) Results: The FeNO was significantly higher in asthmatic children (*n* = 22): 27.3 ± 21.3 ppb; with allergic rhinitis (*n* = 106): 9.9 ± 21.6 ppb, with atopic dermatitis (*n* = 67) 20.8 ± 25.0 ppb, with an asthmatic tendency (*n* = 27): 19.8 ± 16.0 ppb in comparison to children without any respiratory/atopy symptoms. The highest diagnostic odds ratio and area under the curve were found in any treated asthma or asthma without any atopic symptoms in relation to FeNO cutoff > 35 ppb; DOR 4.85 and 8.37; AUC 0.615 and 0.795, respectively. The adjustment for spirometry parameters did not improve the diagnostic accuracy of FeNO. In each FeNO cutoff, there were more false positive than true positive subjects. (5) Conclusions. The best diagnostic accuracy of FeNO was for isolated asthma without any atopy against children without any coexisting respiratory or allergic disease. The sensitivity and specificity did not reach the required values for a good screening tool; therefore, it should not be used in epidemiological settings.

## 1. Introduction

The value of Fractional Exhaled Nitric Oxide measurement (FeNO) in the clinical assessment of asthma is well recognized and currently recommended in establishing the diagnosis, making treatment decisions, and monitoring its effects [1,2,3]. In children suspected of asthma, the measurement of FeNO is recommended if there is diagnostic uncertainty, with the FeNO level of 35 ppb or more considered a positive test of airway inflammation [4]. There is also evidence that a lower level of FeNO (above 25 ppb) is a clinically useful sign of airway inflammation in children with asthmatic symptoms [5]. Less is known about the value of FeNO measurement in epidemiological studies on childhood asthma, although available evidence suggests that an increased FeNO is associated with an increased risk of asthma [6,7]. It remains unclear if a measurement of FeNO can improve screening for asthma in children in the light of the uncertain reliability of questionnaire-derived symptoms, particularly in children. The published evidence concerning the diagnostic value of the FeNO measurement in asthma is more convincing from a clinical point of view and supports the use of the test in predicting asthma [8]. The screening performance of the test in a community setting was addressed by a small number of studies and the results are unequivocal [9,10,11].

The current availability of portable and user-friendly devices measuring FeNO creates possibilities to apply this method in population-based studies on respiratory health in children, including screening for childhood asthma outside of the primary care setting. However, the interpretation of FeNO values obtained through screening must consider a spectrum of conditions leading to an increased FeNO, such as atopy and allergic diseases apart from atopic asthma on one hand, and non-atopic phenotype of asthma on the other hand [12,13]. If proven valid, the FeNO measurement could be used as a supplementary test to identify undiagnosed childhood asthma. Therefore, we designed a study aiming on the assessment of the screening accuracy of FeNO measurement in the identification of children with asthma with participants in a population-based respiratory survey. In the assessment of the screening accuracy, we used a set of diagnostic accuracy measures. The secondary goals of the study were to assess the diagnostic accuracy of FeNO measurements in children with non-asthmatic allergic diseases and to determine if a concomitant measurement of FeNO and spirometric status adds to the identification of asthma in children.

## 2. Materials and Methods

The study subjects were participants (children aged 6–10) of a cross-sectional respiratory health survey conducted between 2017 and 2020. Children were randomly selected from primary schools in four cities (Bytom, Chorzów, Tychy, and Zabrze) located in Silesian Voivodship in Poland. The study protocol included routine epidemiological participation criteria. All children in the selected schools were invited to participate. The children were examined after parental consent and if they qualified for spirometric assessment. The latter criterion followed the ATS/ERS recommendations. Parents or legal guardians of children responded to questions adopted from the International Study of Asthma and Allergies in Childhood (ISAAC) questionnaire [14]. The response rate for the questionnaire was almost 94% (450/480), and one child did not succeed in the FeNO measurement, and 8 children were not able to perform the spirometry test. The following inclusion criteria were considered in the study: 1. Signed informed consent from legal guardian’s child, 2. Completing the adapted ISAAC questionnaire, 3. Relative contraindications for Spirometry from ATS recommendations were excluded [15], 4. The child had no symptoms and had no respiratory infection in the last two weeks. 5. Willing child to participate in the measurements.

Figure 1 presents the process of recruitment for the study. The following respiratory/allergic outcomes were included in the analysis: currently treated asthma, allergic rhinitis, and atopic dermatitis diagnosed by a physician, and asthmatic tendency defined as the attacks of dyspnea and chest wheeze apart from cold during the last 12 months. All diagnoses were based on the ISAAC questionnaire. All asthma cases were identified through the parental answer to the question in a questionnaire: “Has a doctor ever diagnosed asthma in your child?”. Due to the overlap of a respiratory/allergic condition, a subgroup of children was defined. It included children with only one condition without coexisting disorders (only asthma, only allergic rhinitis, only atopic dermatitis, only asthmatic tendency). The FeNO accuracy measures for single conditions were obtained by confronting the performance of the FeNO test in this group and healthy children who had neither out of the abovementioned conditions.

FeNO measurement (electrochemical NIOX MINO device, Circassia, Stockholm, Sweden) was performed between Spring and Autumn seasons, in the middle of the week (Tuesday, Wednesday, or Thursday) and between 8.00 a.m. and 1.00 p.m. via non-nasal airway path. Spirometric tests (Easy One Air, NDD Medizintechnik AG, Zurich, Switzerland) were obtained in the schools, after FeNO measurement. All measurements were performed according to ERS/ATS recommendations [16,17]. Both tests were applied by the trained and certified researcher, during one measurement session. Results of FeNO measurement were expressed in ppb, and three different cutoff values (thresholds) were used in working definitions of increased values of FeNO (“positive cases”): 20 ppb, 25 ppb, and 35 ppb [5,13]. Spirometry included the measurement of forced vital capacity (FVC), forced expiratory volume in the first second (FEV_1_), and the ratio FEV_1_/FVC. All spirometric indices were expressed as absolute values and as a percentage of the predicted values (% p.v.). The values of FVC, FEV_1_ ≥ 80% p.v. and FEV_1_/FVC ≥ 80% were considered normal [18,19].

The complete FeNO and spirometry measurements were obtained in 441 children, one child was unable to finish FeNO and spirometry test and 8 children failed only in spirometry. Informed written consent was obtained from parents or legal guardians of all subjects. The study was conducted in accordance with the Declaration of Helsinki, and the protocol was approved by the Ethics Committee of the Medical University of Silesia in Katowice (decision no KNW/0022/KB1/37/IV/14/16/17/).

### Data Analysis

The quantitative variables were expressed as the arithmetic mean, median, and standard deviation. The qualitative variables were expressed as frequency (*n*) and percentage (%). The Shapiro–Wilk test was used to assess the normal distribution for quantitative variables. The statistical significance differences between distributions of quantitative variables were assessed using the Wilcoxon test. The relationship between qualitative variables was assessed by the Chi-square test or Fisher test, as appropriate, and the association between quantitative variables was assessed using the analysis of correlation modo Spearman.

For three defined thresholds of FeNO values (20, 25, 35 ppb) and in relation to each specific respiratory/allergic outcome the frequency of true positive (TP) and true negative cases (TN), false positive (FP), and false negative (FN) cases was calculated. The set of variables used to assess diagnostic accuracy of FeNO measurement included sensitivity (SENS = TP/(TP + FN), specificity (SPEC = TN/(TN + FP), positive and negative predictive values (PPV = TP/(TP + FP); NPV = TN/(TN + FN), positive and negative likelihood ratios (DLR(+) = SENS/(1 − SPEC); DLR(−) = (1 − SENS)/SPEC), area under the receiver operating characteristic curve (AUROCC) and diagnostic odds ratio (DOR = (SENS/1 − SENS)/1 − SPEC/SPEC) [20,21,22,23]. The effect of FeNO was controlled for age, sex, weight, and height, as well as spirometric variables in simple and multivariate analyses, and no impact of potential confounding factors on FeNO levels was found.

The level of significance in statistical analysis was set at a *p* < 0.05 value. All analyses were performed using SAS statistical package (SAS Institute Inc., Cary, NC, USA, version 9.4).

## 3. Results

The study group included 449 children, 224 (49.9%) boys and 225 (50.1%) girls. The mean age of children was 7.49 ± 0.79 years (median = 7.5 years). Currently treated asthma was found in 22 children (4.9%), 13 boys (6.2%) and 8 girls (3.5%). Allergic rhinitis was diagnosed in 107 children (23.8%), in 61 boys (27.2%) and 46 girls (20.6%). Atopic dermatitis was diagnosed in 69 children (15.3%), in 33 boys (14.7%) and in 36 (16.0%) girls. Asthmatic tendency occurred in 27 children (6.1%), in 17 boys (7.5%), and in 10 girls (4.4%).

The values of FeNO ranged between 5 ppb and 186 ppb with the mean value 15.8 ± 14.8 (median: 12 ppb), 15.6 ± 17.0 (median: 11 ppb) in girls, and 16.1 ± 12.1 (median: 12 ppb) in boys (difference was statistically significant: *p* = 0.04). The values above 20 ppb were found in 83 children (18.5%), the values above 25 ppb were found in 53 children (11.8%), and the values above 35 ppb were found in 29 children (6.5%). Table 1 shows the FeNO values according to four respiratory/allergic outcomes. The largest FeNO values were found in asthmatic children (23.7 ± 21.3 ppb; median = 14 ppb), and in children with the remaining outcomes the mean values ranged between 19.8 ± 16.0 ppb (median = 12 ppb) and 20.8 ± 25.0 ppb (median = 13 ppb). The between-outcome differences in FeNO were not statistically significant. However, except for the group with asthmatic tendency, all other groups defined by outcomes had statistically significantly larger FeNO values than the mean value found in 280 children (mean: 13.7 ± 10.6 ppb; median = 11 ppb) who had neither of the defined respiratory/allergic outcome.

Table 2 shows the frequency of true positives, true negatives, false positives, and false negatives in relation to all three cutoffs of FeNO (>20 ppb, >25 ppb, >30 ppb) for treated asthma, allergic diseases and asthmatic tendency.

Table 3 shows the results of the analysis of the accuracy of the FeNO measurement for all outcomes, according to three different cutoff levels of FeNO performed in the group of all 449 children. In general, the largest area under the curve was obtained in relation to asthma (AUROC = 0.629). AUROC in relation to three other outcomes varied between 0.555 and 0.575. For asthma and depending on the FeNO cutoff, the sensitivity decreased from 32% to 27% and 22%, and the specificity increased from 82% to 89% and 94%. The cutoff of 35 ppb was characterized by the largest positive predictive value (PPV = 17%), positive diagnostic likelihood ratio (DLR+ = 3.98) and diagnostic odds ratio (DOR = 4.85). In comparison to findings in asthma cases for three other outcomes, the same FeNO threshold-related sensitivities were smaller, and specificities were at a similar level. However the largest (at the cutoff = 35 ppb) positive predictive value reached higher levels for allergic rhinitis (PPV = 55%) and atopic dermatitis (PPV = 34%), positive diagnostic likelihood ratio and diagnostic odds ratio were larger for allergic rhinitis (DLR+ = 3.92; DOR = 4.43) than for asthma. Accuracy measures for asthmatic tendency were lower than those for asthma.

The accuracy of FeNO measurement for composed conditions was not better than for single outcomes. When the outcome was defined as any disease diagnosed by a physician (treated asthma or allergic rhinitis or atopic dermatitis found in 158 children) the AUC was 0.58 and DOR ranged from 2.33 to 5.27 with increasing cutoff values of FeNO. When the outcome was composed of either treated asthma or asthmatic tendency, the value of AUROC was 0.58 and DOR varied between 1.98 and 3.62 according to the specific cutoff value.

From 449 children, technically acceptable spirometric measurements were obtained from 350 children. In the whole group, the mean value of FVC was 1.75 ± 0.35 l. (113.5 ± 16.2% p.v.), FEV_1_ 1.50 ± 0.28 l. (105.9 ± 14.6% p.v.) and FEV_1_/FVC 92.2 ± 2.2%. In children with asthma, these values were 1.87 ± 0.42 (122.5 ± 15.4% p.v.), 1.56 ± 0.30 (110.6 ± 8.5% p.v.) and FEV_1_/FVC 92.4 ± 1.6%, respectively. The frequency of abnormal values was for FVC 8.0%, FEV_1_ 10.7% and FEV_1_/FVC 6.1% in the entire group and 1.5% for FVC and FEV_1_, and 5.5% for FEV_1_/FVC. When diagnostic accuracy for asthma was evaluated using both FeNO measurement and spirometric variables, (in% p.v.) in one model, the results did not show any substantial improvement (detailed data not shown). The area under the curve provided by the analysis of asthma regarding FeNO and FEV_1_ increased from 0.61 to 0.66 and even less when other spirometric indices were combined with FeNO. Increases of AUC for other asthmatic tendency in response to FeNO accompanied by FEV_1_ predicted was similarly small, from 0.55 to 0.59.

The study revealed a substantial overlay of the outcomes. For example, out of the 22 cases with asthma, 5 cases were without coexisting allergic disease and multimorbidity in terms of the defined outcomes was also found in relation to allergic rhinitis and atopic dermatitis. To minimize the potentially masking effect of comorbidities on the diagnostic accuracy for a given outcome, the evaluation of accuracy was restricted to the performance of the FeNO test in relation to single outcomes, without coexisting disorders. The restricted subgroup included 22 cases of asthma only (mean FeNO: 23.7 ± 21.3 ppb), 70 cases of allergic rhinitis only (mean FeNO: 17.4 ± 12.9 ppb), 38 cases of atopic dermatitis only (mean FeNO: 17.5 ± 13.9 ppb) and 8 asthmatic tendency only cases (mean FeNO: 19.5 ± 15.1 ppb). Compared with the reference group composed of asymptomatic healthy children, the FeNO levels were statistically significantly higher in children with asthma only (*p* = 0.02), allergic rhinitis only (*p* = 0.04), but not in children with atopic dermatitis (*p* = 0.09) or asthmatic tendency (*p* = 0.2).

While considering diagnostic accuracy estimated for children with single outcomes vis-à-vis the findings in healthy children, compared with the previous estimates of the test performance, the restriction of analysis to single-outcome cases showed the improvement of diagnostic accuracy of FeNO measurement for asthma in terms of the positive diagnostic likelihood ratio (7.95 vs. 3.50 in the non-restricted-group analysis) and diagnostic odds ratio (9.99 vs. 4.08 in the non-restricted-group analysis), with a small increase in area under the curve. The improvement of accuracy for asthmatic tendency was less pronounced and the analogous changes for allergic rhinitis and atopic dermatitis were small.

Within the group with asthma (*n* = 22), 17 children had coexisting allergic rhinitis or atopic dermatitis or both diseases. For this composed outcome, the accuracy of FeNO cutoff at 35 ppb was expressed by sensitivity = 23%, specificity = 97%, positive likelihood ratio = 8.23, diagnostic odds ratio = 10.45 and area under the curve = 0.594.

## 4. Discussion

Our principal goal was to examine not a prognostic value or clinical aspects, but the screening accuracy of FeNO measurement for diagnosis of childhood asthma. In other words, the underlying objective was to screen not for preclinical phase of the disease but for the existing chronic [24,25,26]. While such an application of the measurement of FeNO meets most performance criteria of a screening test (availability, acceptability, substantial morbidity, known treatment), its diagnostic accuracy outside of the clinical setting deserves due assessment [27,28]. In our study, we evaluated the screening performance of FeNO measurement as assessed by the results provided by a set of quantitative indicators of the diagnostic accuracy in relation to asthma and asthma-like respiratory symptoms, against three FeNO thresholds, in a community setting. In addition, the evaluation included the test performance against atopic dermatitis and allergic rhinitis, both known to contribute to the increase in FeNO [29]. It seems that allergic rhinitis plays a key role in subclinical inflammation in the lower airways [30]. The focus on currently treated asthma and not for asthma ever was justified by better reliability of the former manifestation of disease even if diagnosed based on the questionnaire. Moreover, FENO has the best diagnostic accuracy for current asthma [31]. The set of measures of accuracy that was evaluated in our study was composed of both single and aggregated indicators. Most aggregated indices (predictive values, likelihood ratios, and area under the curve) are not independent of the prevalence of the disorder in question, thus we also calculated the diagnostic odds ratio, a single indicator of the test performance that is relatively not prevalence-dependent [24,31,32]. Our findings suggest a low accuracy of the measurement of FeNO in population-based screening for pediatric asthma. For the best cutoff of FeNO level (35 ppb as the level generating the largest sum of sensitivity and specificity), the area under the curve was 0.61, which corresponds with sufficient (AUROC: 0.60–0.69) and not good, very good or excellent performance. The value of the diagnostic odds ratio was small (DOR = 4.85). Moreover, the value of the positive diagnostic likelihood ratio (DLR(+) = 3.98) could be translated to a small probability of the disease being identified by FeN0 above 35 ppb [32]. The measures of accuracy were estimated from the results of analysis including all members of the study group, with no regard to coexisting disorders in asthmatic cases and in the reference group. The restriction of analysis to single diagnostic presentation (asthma only) vis-à-vis healthy controls yielded better measures of accuracy: AUROC = 0.79, DLR(+) = 6.90, DOR = 8.37. However, the asthma-only group included only 5 children. A more realistic evaluation considered 17 cases of asthma coexisting with any diagnosed allergic disease. Unlike the former category of asthma-only, this category could reflect “allergic asthma”, the condition that is more likely to show increased levels of FeNO. For this outcome and for the FeNO cutoff at 35 ppb, the diagnostic accuracy was expressed by AUROC = 5.94, DLR(+) = 8.23 and DOR = 10.45. Because no specific clinical examination was implemented, it remains unclear if asthma-only children had or had not any allergies not reported in the questionnaires.

Our estimates of accuracy can be compared with the published evidence. A comprehensive meta-analysis of studies on the diagnostic accuracy of FENO for discrimination of asthma included three studies in children [33]. However, only two studies focused on respiratory outcomes that corresponded with our study protocol, although the age span of subjects was wider than that in our study [10,34]. The Korean study evaluated the utility of FeNO measurement using the clinical diagnosis of asthma. The best cutoff FeNO value of 22 ppb was associated with 56% sensitivity and 87% specificity. Both indices were larger when atopic asthma was considered (72% and 85%, respectively) [12]. The study performed in Israel provided evidence that FeNO measurement is useful in the early diagnosis of pediatric asthma [34]. The sensitivity, specificity, positive and negative predictive values for the best cutoff of FeNO (19 ppb) were 80%, 92%, 89% and 86%, respectively, and the area under the curve was 0.90. The results were obtained in 106 children with clinically confirmed asthma and in 44 children without asthma. The published findings, unlike our results, are in favor of the diagnostic utility of the FeNO measurement. It cannot be excluded that the difference between our findings and published evidence is related to the difference of the composition of the group and in the case definition. Moreover, apart from relying on the clinical diagnosis of asthma, both studies included selected children, suspected of asthma, and a study from Israel was a prospective observation. On the contrary, the conclusions, like our findings, were reported by the authors of two large population-based studies involving questionnaires as the tool to identify asthma [35,36]. Their major findings can be summarized by the small area under the curve, 0.53 and <0.70, respectively. Moreover, the authors of the former study concluded that FeNO measurement could be used not to identify asthma but to exclude the diagnosis in schoolchildren, and the authors of the latter study did not recommend FeNO measurement for identifying pediatric asthma in a community setting. It is also of interest that our estimates of accuracy were similar to the results obtained in adolescents, participants of a population-based birth cohort study. The study showed that for the FeNO cutoff of 35 ppb and in subjects reporting one or more respiratory symptoms and diagnosed with asthma, AUROC was 0.62, with 44% sensitivity, 84% specificity [31]. An earlier study performed in England yielded results that are also in line with our findings. That study included 368 schoolchildren and showed that increased levels of FeNO did not discriminate between children with asthmatic and atopic symptoms, although the FeNO levels were higher in those groups than in healthy control children [11]. The conclusion was based on the results of comparisons of FeNO distributions between atopic asthmatic, non-atopic asthmatic, atopic only and healthy children in a community setting. The disorders were diagnosed using questionnaires and family physician’s notes, and no indices of diagnostic accuracy of FeNO cutoffs were calculated.

In our study, FeNO levels were higher in allergic rhinitis and atopic dermatitis than in healthy controls and lower than in asthmatic children, and this finding is not surprising in the light of the published evidence [29,37]. Compared with the screening accuracy for asthma, the performance of the FeNO test in relation to allergic rhinitis and atopic dermatitis was lower and could be classified by area under the curve as not sufficient. Moreover, the measurement of FeNO appeared to have no practical value in identifying children with asthmatic tendency; it is probably due to the fact that there are many undiscovered factors that influence FeNO [38]. The combination of the FeNO measurement with spirometry did not improve the diagnostic accuracy for the defined respiratory outcomes and this finding corresponds with the view of a low diagnostic value of spirometric assessment in asthma [31]. The strengths of our study stem from its population-based design. We evaluated the screening performance of FeNO measurement for asthma in a real-life scenario and we addressed the diagnosed cases of currently treated asthma and not asthma ever diagnosed. Moreover, the results of comparison of the screening performance of FeNO measurement for asthma (higher accuracy) with allergic rhinitis and atopic dermatitis (lower accuracy) add credibility to our findings. Nonetheless, the apparent limitation of our study arises from the definition of asthma. Our principal questionnaire-based outcome was asthma under current treatment, the item frequently used in epidemiological studies [39]. However, without clinical assessment, the accuracy of any questionnaire-based definition of asthma remains unknown [40]. Similar uncertainty relates to other questionnaire-based respiratory outcomes and both reservations should be considered in the interpretation of the predictive value of FeNO measurement in screening for chronic respiratory conditions in children, including asthma. Another factor hampering interpretation is the lack of convincing distinction between eosinophilic and non-eosinophilic asthma. The same situation was related to the assessment of atopy/non-atopy status. The children were diagnosed only on the answer to the question in the questionnaire: “Has a physician ever diagnosed rhinitis in your child?”. However, a clinical study showed that the prognostic value of FeNO measurement for asthma diagnosis was better for a delayed (12 month) onset of eosinophilic asthma (AUC = 0.82) than for all diagnoses of asthma (AUROC = 0.60) [41]. This shortcoming can be however lessened by the finding that, in our study, the largest accuracy for asthma is with coexisting allergic disorders, and this is in line with the pathomechanism of FeNO synthesis. Another potentially masking factor is the suppression of FeNO levels by treatment with steroids, the issue not addressed in our study [4]. Another limitation of our study is that our study protocol did not allow insight into the use of corticosteroids in asthmatic children. Inhaled corticosteroids are known to reduce airway inflammation and FeNO synthesis as the result.

## 5. Conclusions 

In a real-life scenario, and especially for the asthma screening purposes, the issue of disorders coexisting with asthma is a secondary question. The principal goal is to identify any asthma, with or without comorbidities. From this point of view, and for a given population, a low sensitivity and low specificity play a decisive role. In our study and for the best cutoff of FeNO, both indices translate to a large fraction of false positive results among all positive results of the screening test (24/29 = 82%), even if the overall frequency of all positives is low (6.4%). Many false positive results trigger an unnecessary burden of diagnostic procedures and can cause unwanted psychological problems such as anxiety in the whole family of a screened child [42]. Several false negative cases, larger than the number of true positive cases in our study (17 vs. 5), cannot be neglected either. As a result, our findings do not support the view that FeNO measurement is an effective screening test for pediatric asthma in a community setting.

## Figures and Tables

**Figure 1 children-09-00858-f001:**
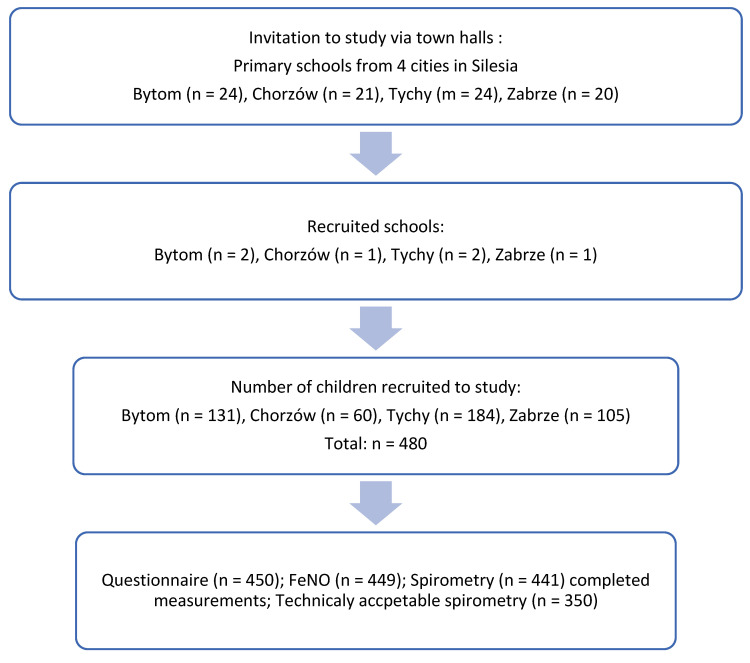
The process of recruitment of children in the study.

**Table 1 children-09-00858-t001:** Mean values of FeNO according to the respiratory/allergic outcome.

Outcome	*n*	Mean ± SD	Median	*p*-Value *
Asthma	22	27.3 ± 21.3 ppb	14 ppb	0.02
Allergic rhinitis	107	19.9 ± 21.6 ppb	13 ppb	0.01
Atopic dermatitis	69	20.8 ± 25.0 ppb	13 ppb	0.01
Asthmatic tendency	27	19.8 ± 16.0 ppb	12 ppb	0.1

Legend: *—statistical significance of the difference between the value in children with a given outcome compared with 280 children without any outcome (mean: 13.7 ± 10.6 ppb; median = 11 ppb).

**Table 2 children-09-00858-t002:** Frequency (*n*) of true positives (TP), true negatives (TN), false positives (FP), and false negatives (FN) of three FeNO cutoffs for the study outcomes.

Outcome	FeNO [ppb]	TP [*n*]	TN [*n*]	FP [*n*]	FN [*n*]
Asthma *n* = 22	>20	7	351	76	15
	>25	6	380	47	16
	>35	5	413	24	17
Allergic rhinitis *n* = 107	>20	30	289	53	77
	>25	23	312	30	84
	>35	16	329	13	91
Atopic dermatitis *n* = 69	>20	20	317	63	49
	>25	15	342	38	54
	>35	10	361	19	59
Asthmatic tendency *n* = 27	>20	8	347	75	19
	>25	7	376	46	20
	>35	5	398	24	22

**Table 3 children-09-00858-t003:** Diagnostic accuracy (sensitivity, specificity, positive and negative predictive value, positive and negative diagnostic likelihood ratio, diagnostic odds ratio, and area under curve) of FeNO in detecting treated asthma, allergic rhinitis, atopic dermatitis, and asthmatic tendency in all children.

Outcome	FeNO [ppb]	SEN	SPE	PPV	NPV	DLR(+)	DLR(−)	DOR	AUROC
Asthma *n* = 22	>20	31.8	82.2	8.4	95.9	1.79	0.83	2.15	0.615
	>25	27.2	88.9	12.3	95.9	2.45	0.82	2.36	
	>35	22.7	94.3	17.2	95.9	3.98	0.86	4.85	
Allergic rhinitis *n* = 107	>20	28.0	84.5	36.1	78.9	1.81	0.85	2.12	0.568
	>25	21.5	91.2	43.4	78.8	2.44	0.86	2.83	
	>35	14.9	96.2	55.1	78.3	3.92	0.88	4.43	
Atopic dermatitis *n* = 69	>20	28.9	83.4	24.1	86.6	1.74	0.85	2.04	0.575
	>25	21.7	90.0	28.3	86.3	2.17	0.87	2.49	
	>35	14.4	95.0	34.4	85.9	2.88	0.90	3.19	
Asthmatic tendency *n* = 27	>20	29.6	82.2	9.6	94.8	1.66	0.86	1.94	0.555
	>25	25.9	89.1	13.2	94.9	2.38	0.83	2.85	
	>35	18.5	94.3	17.2	94.7	3.25	0.86	3.75	

Legend: SEN—sensitivity, SPE—specificity, PPV—positive predictive value, NPV—negative predictive value, DLR(+)—positive diagnostic likelihood ratio (positive), DLR(−)—negative diagnostic likelihood ratio (positive), DOR—diagnostic odd ratio, AUROC—area under curve provided by receiver operating characteristic.

## Data Availability

The data are available on a reasonable request from the corresponding author.

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
