# Peer review of "Screening Accuracy of FeNO Measurement for Childhood Asthma in a Community Setting"

_children, 2022, doi:10.3390/children9060858_

Round 1
Reviewer 1 Report
Indeed, the utility of FeNO in asthma diagnosis and management is debatable and a topic of great interest. This manuscript was well-written and the findings add significant data to the existing literature. The authors are to be commended.
One point to be noted when referring to spirometry parameters ( FEV1, FEV1/FVC etc), the '1' is to be written as a subscript.
No further comments.
Author Response
Dear Reviewer 1,
thank you for your time.
We changed all ‘1’ in spirometry parameters for subscript as recommended.
Kindly see, the tracking changes.
Moreover, the manuscript was improved by the native speaker.
Kind regards,
Authors
Reviewer 2 Report
The article „Screening accuracy of FeNO measurement for childhood 2 asthma in a community setting” highlights the role of FENO as a screening method for asthma in children. The authors also highlighted possible recommendations in the FENO cut-off values, for asthma screening. I have read the paper with interest and feel that it is relevant for area of Asthma diagnosis.
I suggest few major revisions and comments are made below regarding the article.
- The authors should clarify the inclusion/exclusion criteria: mainly exclusion of respiratory infections in the time of FENO measurements.
- In the Methods: the diagnosis of Allergic Asthma is made using the ISAAC questionnaire, please ad limitation regarding the atopic/non-atopic disease.
- Describe the moment of the FENO measurement towards allergen exposure, because for example: pollen allergic patients outside the pollen season may have normal nasal FENO values.
- In Discussion: Add discussion regarding the influence of Allergic Rhinitis in underling subclinical inflammation in lower airways.
Please read and cite article: “Muntean IA, Bocsan IC, Vesa S, Miron N, Nedelea I, Buzoianu AD, Deleanu D. Could FeNO Predict Asthma in Patients with House Dust Mites Allergic Rhinitis? Medicina (Kaunas). 2020 May 14;56(5):235. doi: 10.3390/medicina56050235. PMID: 32422966; PMCID: PMC7279291.”
- Please add “Limitations of the study” in discussion. Please revise.
- References should be modified according to the journal requests for publication

Author Response
Dear Reviewer,
thank you for the effort that your put into increasing the value of the manuscript.
The first comment is related to inclusion/exclusion criteria. The reviewer wrote:
- The authors should clarify the inclusion/exclusion criteria: mainly exclusion of respiratory infections in the time of FENO measurements.
In the Materials and Methods we added the inclusion criteria
The following inclusion criteria were considered in the study: 1. Signed informed con-sent from legal guardian’s child, 2. Fulfilling the adaptation of ISAAC questionnaire, 3. Relative contraindications for Spirometry from ATS recommendations [16], 4. Willing child to participate in the measurements.
- In the Methods: the diagnosis of Allergic Asthma is made using the ISAAC questionnaire, please ad limitation regarding the atopic/non-atopic disease.
The diagnosis of allergic asthma was made only on ISAAC questionnaire, the assessment of the status of atopic or non-atopic status of the disease was assessed according to further questions such as: has physician ever diagnosed rhinitis in your child?
We added sentence in the discussion section: The same situation was related to assessment of atopy/non-atopy status. The children were diagnosed only on the answer for the question in the questionnaire: “has physician ever diagnosed rhinitis in your child?”. Lines 325-328.
- Describe the moment of the FENO measurement towards allergen exposure, because for example: pollen allergic patients outside the pollen season may have normal nasal FENO values.
We added The FeNO measurement was performed indoor during spring season via non-nasal airway path. FeNO measurement (electrochemical NIOX MINO device, Circassia, Stockholm, Sweden) was performed between Spring and Autumn session, in the middle of the week (Thuesday, Wednesday or Thurdsay) via non-nasal airway path., followed by sSpirometric tests (Easy One Air, NDD Medizintechnik AG, Zurich, Switzerland) were obtained in the schools, after FeNO measurement.
- In Discussion: Add discussion regarding the influence of Allergic Rhinitis in underling subclinical inflammation in lower airways.
We added information about allergic rhinitis regarding the subclinical inflammation in lower airways and cited the suggested article. Thank you.
Please read and cite article: “Muntean IA, Bocsan IC, Vesa S, Miron N, Nedelea I, Buzoianu AD, Deleanu D. Could FeNO Predict Asthma in Patients with House Dust Mites Allergic Rhinitis? Medicina (Kaunas). 2020 May 14;56(5):235. doi: 10.3390/medicina56050235. PMID: 32422966; PMCID: PMC7279291.”
- Please add “Limitations of the study” in discussion. Please revise.
The limitation of the study are involved in the discussion. Kindly see lines 319 and further. Thank you.
- References should be modified according to the journal requests for publication
Done, as recommended. Thank you
Moreover, the manuscript is improved by the native speakers.
Kind regards,
Authors
Reviewer 3 Report
This cross-sectional study examined the screening accuracy of FeNO measurement for childhood asthma in a community setting. The result of the study may have a clinical implication but there is a major concern. In the analysis, several important factors that may affect FeNO levels were not considered, including age, sex, height, time of measurement (morning or afternoon), use of medication (inhaled corticosteroid or other anti-inflammatory drugs), airway infection in the past 2 weeks, exposure to second-hand smoking. Were these information available? Study design with FeNO should consider to control some of these characteristics.
Author Response
Dear Reviewer, thank you for your valuable comment. We updated the manuscript according to your recommendations.
We performed a very deep analysis of our data and did control the potential confounding factors: lines: 131-132. The effect of FeNO was controlled for age, sex, weight and height as well as spirometric variables in simple and multivariate analyses and we did not find any impact of potential confounding factors on FeNO levels.
As well, we mentioned the impact of treatment in the discussion section. The second hand-smoking was controlled via methodology (middle of the week) and statistical analysis. From our previous studies there were no impact of second
As well, we expanded the information about FeNO measurements.
Moreover, the manuscript is improved by the native speaker.
Kind regards,
Authors
Round 2
Reviewer 2 Report
The article „Screening accuracy of FeNO measurement for childhood 2 asthma in a community setting” highlights the role of FENO as a screening method for asthma in children. The authors also highlighted possible recommendations in the FENO cut-off values, for asthma screening. I have read the paper with interest and feel that it is relevant for area of Asthma diagnosis. I suggest few minor revisions regarding English language style and punctuation.
Author Response
Dear Reviewer, thank you for time and effort that you put improve our manuscript. The language style and punctuation have been corrected.
Regards,
Authors
Reviewer 3 Report
I thank for the authors to make changes to the article. The manuscript has been improved and has responded the comments from reviewers.
Author Response
Dear Reviewer, thank you for time and effort that you put to improve our manuscript. The language style and punctuation have been corrected.
Regards,
Authors